# Optimising PHBV biopolymer production in haloarchaea via CRISPRi-mediated redirection of carbon flux

Lin Lin[1,2], Junyu Chen[1], Ruchira Mitra[1,3], Quanxiu Gao[1], Feiyue Cheng[1,2], Tong Xu[1], Zhenqiang Zuo[1], Hua Xiang [1,2✉] & Jing Han [1,2✉]

The haloarchaeon *Haloferax mediterranei* is a potential strain for poly(3-hydroxybutyrate-*co*-3-hydroxyvalerate) (PHBV) production, yet the production yield and cost are the major obstacles hindering the use of this archaeal strain. Leveraging the endogenous type I-B CRISPR-Cas system in *H. mediterranei*, we develop a CRISPR-based interference (CRISPRi) approach that allows to regulate the metabolic pathways related to PHBV synthesis, thereby enhancing PHBV production. Our CRISPRi approach can downregulate the gene expression in a range of 25% to 98% depending upon the target region. Importantly, plasmid-mediated CRISPRi downregulation on the citrate synthase genes (*citZ* and *gltA*) improves the PHBV accumulation by 76.4% (from 1.78 to 3.14 g/L). When crRNA cassette integrated into chromosome, this further shortens the PHBV fermentation period and enhances PHA productivity by 165%. Our transcriptome analysis shows that repression of citrate synthase genes redirects metabolic flux from the central metabolic pathways to PHBV synthesis pathway. These findings demonstrate that the CRISPRi-based gene regulation is a transformative toolkit for fine-tuning the endogenous metabolic pathways in the archaeal system, which can be applied to not only the biopolymer production but also many other applications.

[1] State Key Laboratory of Microbial Resources, Institute of Microbiology, Chinese Academy of Sciences, Beijing, People's Republic of China. [2] College of Life Science, University of Chinese Academy of Sciences, Beijing, People's Republic of China. [3] International College, University of Chinese Academy of Sciences, Beijing, People's Republic of China. ✉email: xiangh@im.ac.cn; hanjing@im.ac.cn

Halophilic archaea (haloarchaea) are a distinct branch of the Archaea domain that inhabits the high salinity environments[1]. *Haloferax mediterranei* is the most common model strain for studying haloarchaeal physiology and metabolism. Compared to the other known haloarchaea, *H. mediterranei* possesses several advantages. It grows fast and uses a wide range of carbon sources, including simple sugars, and low-cost substrates such as agro-industrial wastes[2]. The high-salt concentration required for the growth of *H. mediterranei* reduces the chances of microbial contamination and eases sterilization procedures during fermentation[3]. Furthermore, the hyper salinity creates high intracellular osmotic pressure on water exposure, which leads to easy lysis of *H. mediterranei* cells, hence simplifying downstream processing[4]. Importantly, *H. mediterranei* is a potential producer of industrially valuable products such as carotenoid[5] and especially, poly(3-hydroxybutyrate-*co*-3-hydroxyvalerate) (PHBV)[6].

Polyhydroxyalkanoates (PHAs) are bio-polyesters synthesized by many bacteria and archaea as carbon and energy storage usually under nutrient-limiting conditions[7]. These biodegradable and biocompatible biopolyesters have been drawing much attention as environmentally friendly bioplastics and biomedical materials, due to their excellent material properties[8,9]. The physico-chemical and biological properties of these biopolymers can be enhanced following a wide range of developed strategies for a great number of industrial applications[10,11]. Poly(3-hydroxybutyrate) (PHB) and PHBV are two extensively researched PHAs. Compared to PHB, PHBV is more promising for industrial and biomedical applications[12]. PHBV is less crystalline, more flexible and exhibits high processability. Thus, it is gaining increasing importance in various biomedical applications like tissue engineering scaffold fabrication, wound healing, and medical implant development. However, most bacteria and archaea synthesize PHB but are incapable of producing PHBV naturally[13]. They require high-priced 3HV precursor supplementation that increases PHBV production cost. *H. mediterranei* is one among the very few microorganisms that can synthesize PHBV from simple and cheap substrates, without any 3HV precursor supplementation[14].

At present, the main limitation of PHBV application is its high production cost. In order to increase the production of PHBV, a series of strategies have been adopted, such as optimizing culture conditions and feeding strategies, and using low cost agro-industrial wastes as substrate[15–17]. Besides, an excellent chassis cell is extremely important for the improvement of PHBV production. Genetic manipulation is a necessary means to obtain excellent chassis cells. In our previous work, exopolysaccharides (EPS) gene cluster in *H. mediterranei* was knockout to improve PHBV production by 20%[18]. Another engineered strain, with EPS gene cluster and phosphoenolpyruvate synthetase-like gene knockout, accumulated even higher amounts of PHBV[4]. Developing a further better production platform based on *H. mediterranei* is still ongoing and thus, more synthetic biology researches need to be conducted. Central metabolism is the main destination of carbon flux. It is difficult to direct carbon flux from central metabolism to PHBV synthesis by traditional genetic manipulation methods such as gene knockout, because that needs to deal with essential genes. In this context, a sequence-specific regulation system for gene expression is in an urgent need. Recently, CRISPRi (<u>c</u>lustered <u>r</u>egularly <u>i</u>nterspaced <u>s</u>hort <u>p</u>alindromic <u>r</u>epeats <u>i</u>nterference) is gaining considerable interests as a powerful tool for the repression of gene expression in eukaryotes and bacteria[19,20]. Using this technique, essential genes are speculated to be repressed to various degrees and their influence on PHBV synthesis can be determined.

CRISPR-Cas is an RNA-guided adaptive immune system of most archaea and bacteria that provides sequence-specific protection to host against foreign genetic elements[21,22]. As early as 1993, sequences similar to CRISPR loci were found in *H. mediterranei*[23]. Later, our research group reported the subtype I-B CRISPR-Cas system of this strain[24]. It consists of eight Cas proteins and six CRISPR loci (four arrays located on the chromosome, and two arrays located on the pHM500 megaplasmid) that are actively expressed. Typically, in type I-B CRISPR-Cas system, Cas6 recruits Cas5, Cas7, and Cas8 to form Cascade, which is guided by a crRNA to bind to a target DNA sequence[25]. A prerequisite for this binding process is the presence of the protospacer adjacent motif (PAM) directly adjacent to the target sequence[26]. After binding of Cascade, Cas3 is recruited to degrade the target DNA. If *cas3* is deleted, the target DNA would not be degraded, and the DNA-binding function of Cascade will be retained[27]. Subsequently, Cascade binding to the promoter region or the open reading frame of a gene will block the binding or the transcription process of RNAP (RNA polymerase), and thus inhibit transcription initiation or elongation. Based on this principle, the endogenous type I-B CRISPR-Cas system can be developed into CRISPRi tool for targeted gene regulation.

CRISPRi tools have been developed and applied in several bacterial genera and eukaryotes for precise regulation of their gene expression, and further modulation of the metabolic flux[28,29]. However, it has been much less explored in archaea. The first CRISPRi of archaeal organisms was developed in the haloarchaeon *Haloferax volcanii*[30,31]. However, there are no CRISPRi applications to metabolic engineering or synthetic biology in archaea. As evidenced by the CRISPRi-mediated metabolic engineering of bacteria, this technology offers a new approach to enhance PHBV production. Acetyl-CoA is an important precursor for both TCA cycle and PHB or PHBV biosynthesis[32]. Citrate synthase is an essential enzyme that catalyzes the first reaction of the TCA cycle by condensing acetyl-CoA and oxaloacetate to generate citrate[33]. It is expected that repression of the citrate synthase gene would channelize more acetyl-CoA towards PHAs biosynthesis. In *E. coli*[32] and *Halomonas* species TD01[34], this gene was partially repressed using CRISPRi system and PHB synthesis was improved. Thus, CRISPRi is a specific and efficient tool that can be employed to downregulate the expression of essential genes. It is an innovative platform that has revolutionized synthetic biology strategies in microbial strains.

In the present study, CRISPRi technology has been implemented in archaeal system to improve PHBV biosynthesis. CRISPRi tool was developed based on the endogenous type I-B CRISPR-Cas system in *H. mediterranei*. In a nutshell, this study aims to implement CRISPRi as an efficient tool for metabolic engineering in archaea towards improved production of high value-added PHBV.

## Results

It is difficult to apply the traditional CRISPR-dCas9 system in *H. mediterranei* for gene repression due to the high intracellular salt concentration. Thus, the endogenous CRISPR-Cas system was used to develop the CRISPRi tool in this strain (Fig. 1a, b). The identification of PAM sequences is a prerequisite for the development of CRISPR-Cas-based gene editing tools. Therefore, in this study, we first identified the functional PAM sequence required for CRISPR interference in *H. mediterranei*. Next, *cas3* was knocked out to prevent the Cas3 protein from degrading target DNA. Then, different crRNAs were designed to target different regions of the key gene *crtI* involved in lycopene synthesis. The inhibitory effects generated by different crRNAs

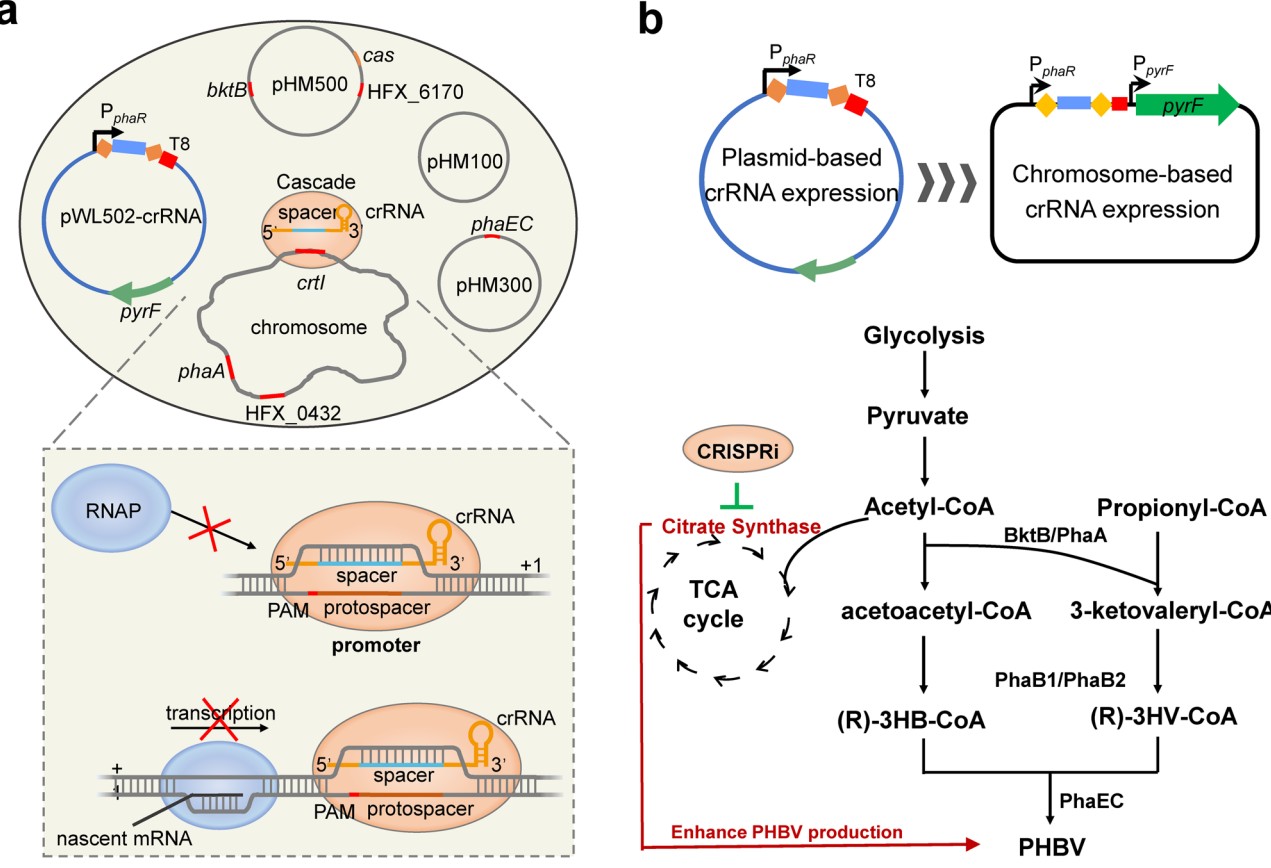

**Fig. 1 Schematic illustration of CRISPRi technology and its application for enhancing PHBV production in _H. mediterranei_. a** Cascade guided by a crRNA targeting the promoter region of a gene blocks access of RNAP and thereby inhibits transcription initiation. Cascade mediated by a crRNA that binds within the open reading frame of a gene prevents RNAP from proceeding along the gene and thus stops transcription elongation. CRISPRi is mediated by expressing crRNAs using derivative plasmids of pWL502 carrying a _phaR_ promoter, repeats (orange diamonds), different spacers (light blue) and a T8 terminator (red rectangle). **b** crRNA expression cassette is integrated into the original locus of _pyrF_ gene in the genome. CRISPRi is used to enhance PHBV production in _H. mediterranei_ by repressing citrate synthase genes. BktB/PhaA, β-ketothiolase; PhaB1/PhaB2, acetoacetyl-CoA reductase; PhaEC, PHBV synthase.

were determined. After the development of CRISPRi, the repression effects on gene expression were evaluated systematically, including different types of genes, simultaneous repression of two genes/gene clusters, and the inhibitory effect of a gene in different growth periods. We subsequently applied plasmid-based CRISPRi to repress citrate synthase genes to enhance PHBV synthesis in _H. mediterranei_. Finally, the crRNA expression cassette integrated into _H. mediterranei_ genome further improved PHBV productivity.

**TTG and TTC are two functional PAMs.** Plasmid-based invader assay was performed to determine the PAM sequence required for CRISPR interference in _H. mediterranei_. Based on the previous studies of PAM sequences in haloarchaea with type I-B CRISPR-Cas system, this motif consists of three nucleotides and located upstream of protospacers[35,36]. Usually, PAM sequence depends on the type of CRISPR-Cas system, CRISPR repeat cluster and microorganism species[37]. Therefore, we chose the functional PAMs recognized for CRISPR interference in two different haloarchaea as our candidate PAMs, which included four functional PAMs (TTC, TTG, TTT, and CCC) in _Haloarcula hispanica_[35] and four functional PAMs (ACT, TAG, TAT and CAC) in _H. volcanii_[36]. Considering the high GC content of the _H. mediterranei_ genome, PAM sequences with a high GC content might have higher probability of targeting a greater number of

genes. Thus, we simultaneously tested another seven PAM sequences with a high GC ratio, namely AGG, ACC, CCT, CGA, CGG, GCT, and TGC.

A sequence matching to the spacer1 of CRISPR locus P23 (P23-S1, located on the pHM300 megaplasmid) was used as the protospacer to construct invader plasmids. A total of fifteen plasmids named pWL502-NNN (Supplementary Data 1, NNN represents the distinct PAM sequence) bearing protospacer sequences and candidate PAMs were constructed. DF50ΔEPS, a _pyrF_-deficient strain unable to grow without uracil supplementation, was used as the host for invader plasmid transformation (Fig. 2a). Plasmids bearing protospacer sequences with a functional PAM efficiently triggered CRISPR-mediated defense in cells carrying the corresponding spacer sequence[36]. These plasmids were recognized and then degraded by the native CRISPR-Cas system. As a result, such cells failed to grow without uracil. On the other hand, the invader plasmids with a non-functional PAM were retained in the transformants, because it couldn't trigger the CRISPR-Cas interference response. Thus, these transformants continued to grow without requiring uracil supplementation (Fig. 2a).

Various transformation rates of invader plasmids were observed (Fig. 2b), suggesting that interference activity was affected by different PAM sequences. Among the fifteen PAM constructs tested, the plasmids pWL502-TTG and pWL502-TTC showed an obvious reduction in transformation rate, indicating

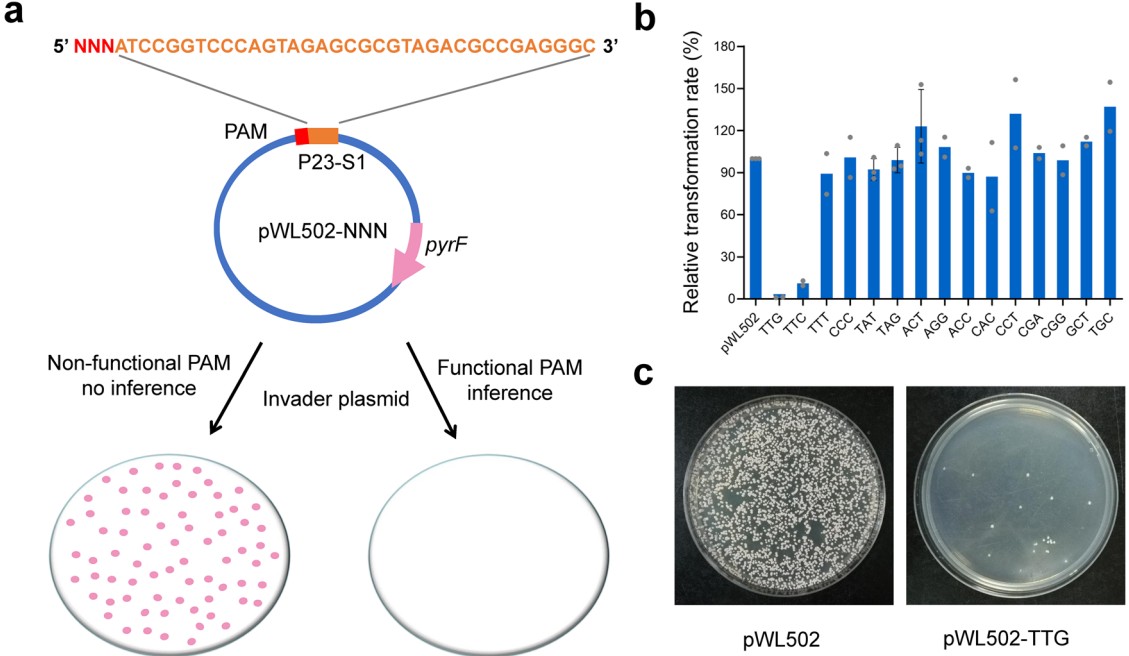

**Fig. 2 Plasmid-based invader assay for identification of functional PAMs. a** The schematic representation of the plasmid-based invader assay. Potential PAM sequences (NNN) are introduced upstream of the spacer sequence P23-S1 (orange letters) and then inserted into pWL502 yielding invader plasmids pWL502-NNN. DF50ΔEPS is transformed with invader plasmids and selected on AS-168SY plates without uracil. Only the cells containing the invader plasmid with a non-functional PAM could grow on the plates. No cell growth indicates that the invader plasmid carries a functional PAM which leads to interference and degradation of the invader plasmid. **b** Relative transformation rates of PAM constructs. Transformation rates are calculated as the number of transformants/μg DNA. The transformation rate of pWL502 is used as the positive control. Data shown for two or three biological replicates. Error bars indicate SDs, n = 3. **c** Colonies obtained by transformation of 1 μg of plasmid pWL502 or pWL502-TTG.

that TTG and TTC were two functional PAMs. Notably, the relative transformation rate of pWL502-TTG was the lowest (1.1%), resulting in very few transformants on the plate (Fig. 2c). These escaped mutants survived from the plasmid challenge possibly by generating mutation or deletion of the PAM and/or spacer P23-S1 and/or *cas* genes. Transformation rates of the rest thirteen PAM constructs were similar to that of control plasmid pWL502, indicating that they were non-functional PAM constructs. TTG was chosen as the efficient PAM sequence for our subsequent experiments, because it resulted in the best interference effect among the fifteen PAMs tested.

**crRNAs targeting the promoter region have the highest effect.** The type I-B CRISPR-Cas system employs Cas3 as the genetic scissor to cleave the target DNA sequence. Therefore, in order to develop a CRISPRi tool using this endogenous system, we first deleted *cas3* to inactivate the cleavage ability in the lycopene producing strain 50BΔ2549, generating 50BΔ2549Δcas3. Reportedly, the binding site of a crRNA influences the repressive effect[19,31]. Thus, crRNAs targeting different regions of *crtI* (HFX_2550) were designed and expressed based on plasmid pWL502 in 50BΔ2549Δcas3 to study their inhibitory effects. *crtI* is the coding gene of phytoene desaturase that catalyzes the formation of lycopene from phytoene. The strain 50BΔ2549Δcas3 produced lycopene and thus exhibited orange color. Deletion of *crtI* changed the cell color from orange to white. Repression of *crtI* in 50BΔ2549Δcas3 was speculated to reduce lycopene production and thus lighten the cell color. Therefore, the change in cell color served as an indicator of *crtI* repression in this strain.

In order to analyze the binding regions of crRNAs, CR-RT-PCR (Supplementary Methods) was used to identify the transcription start site (TSS) and approximate range of the *crtI* promoter. It showed that *crtI* was transcribed from the A of the

start codon ATG (Fig. 3a), which is a typical characteristic for leaderless transcripts in haloarchaea. Six crRNAs targeting different regions of *crtI* were designed according to the position of the PAM sequence TTG (Fig. 3a). Among them, three crRNAs, crtI-t1, t2, and t3, were designed to target the template strand of *crtI*, while crtI-c1, c2, and c3 targeted the coding strand of *crtI*. t1 and c1 targeted the promoter region of *crtI*, while the remaining four crRNAs targeted different regions of the open reading frame. The strain 50BΔ2549Δcas3 was transformed with CRISPR-carrying plasmids which expressed the six crRNAs against *crtI*, respectively. The strains of 50BΔ2549Δcas3 and 50BΔ2549ΔcrtI carrying pWL502 were used as controls. When cultured in MG medium, the transformants appeared in various shades of white to orange (Fig. 3b), showing that *crtI* was inhibited to varying degrees by CRISPRi. Among the six tested crRNAs, crtI-c1 had the most obvious effect on color change. The color of the strain expressing crtI-c1 was almost similar to the white color of the control strain 50BΔ2549ΔcrtI, indicating an obvious reduction in lycopene production and *crtI* gene expression. In addition to crtI-c1, the color change of the strain expressing crtI-t1 was also very obvious, and it led to the second most effective repression after crtI-c1. crtI-c2 lightened the cell color slightly while no obvious color change of the strains expressing crtI-c3, t2, and t3 was observed.

To further quantify the change in lycopene production and gene expression, lycopene quantification (Fig. 3c) and RT-qPCR analysis (Fig. 3d) were performed, respectively. In accordance with our observation, the white strain expressing crtI-c1 showed the lowest level of lycopene production and *crtI* gene expression. crtI-c1 expression reduced lycopene production and the *crtI* transcriptional level down to 2%. This repression effect was comparable to that of *crtI* deletion. The strain expressing crtI-t1 showed the second-lowest lycopene production and gene expression. It resulted

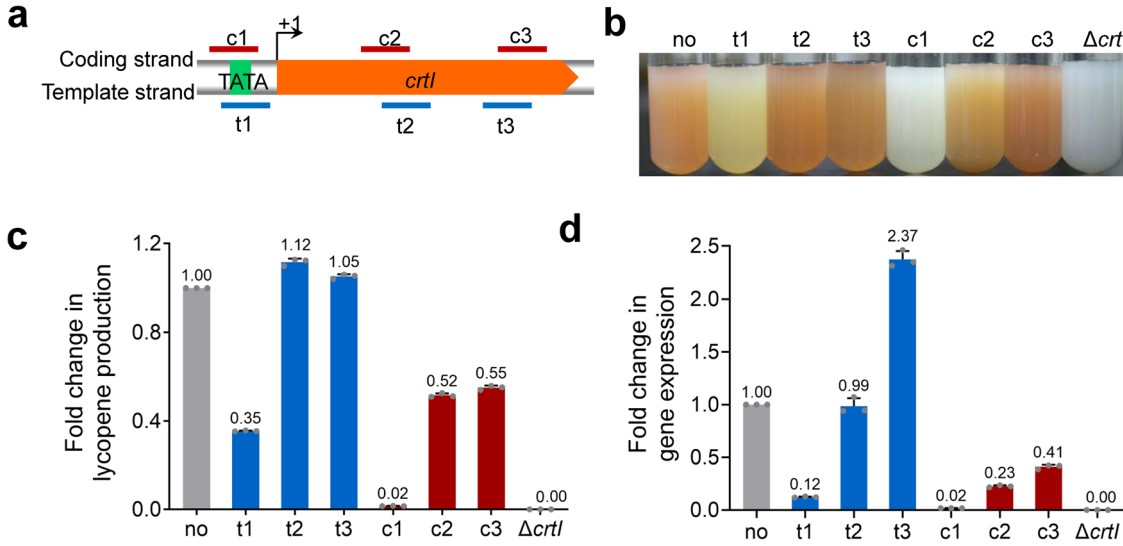

**Fig. 3 Repression of *crtI* gene using CRISPRi. a** Location of crRNA target sites in the *crtI* gene. *crtI* is transcribed from the A of the start codon ATG. The TSS is shown as +1. t1, t2, and t3 are crRNAs targeting the template strand of *crtI*. c1, c2, and c3 are crRNAs targeting the coding strand of *crtI*. **b** Cell colors of 50BΔ2549Δcas3 expressing different crRNAs targeting *crtI*. **c** Fold change in lycopene production of 50BΔ2549Δcas3 expressing different crRNAs targeting *crtI*. **d** Fold change in the expression of *crtI*. In **b**, **c**, **d** no and Δ*crtI* represents the control strain with no crRNA expression and with *crtI* knockout, respectively. Data shown for three biological replicates. Error bars indicate SDs, *n* = 3.

in a reduction of lycopene production down to 35% and mRNA level down to 12%. In addition, the cells expressing crtI-c2 or c3 had a slight change in color, with lycopene production reduced down to 52% and 55%, respectively. Meanwhile, the respective expression of *crtI* was reduced down to 23 and 41%. However, there was no noticeable reduction in lycopene production and gene expression in cells expressing crtI-t2 or t3. Overall, the four crRNAs, crtI-t1, c1, c2, and c3, differentially downregulated *crtI* expression by targeting different regions of the template strand and coding strand of *crtI* gene. However, strong repression effect was achieved by expressing crtI-c1 and t1, which targeted the promoter region indicating that binding of crRNAs to the promoter region of either template strand or coding strand facilitated gene repression most efficiently. Our results indicated that the CRISPRi system could be used to achieve different levels of gene regulation in *H. mediterranei*.

**CRISPRi repressed different types of genes effectively**. It has been shown that CRISPRi can repress the expression of a non-essential chromosomal gene *crtI* efficiently in *H. mediterranei*. We next studied the repression effect of the CRISPRi system on essential genes, megaplasmid genes and gene clusters. Based on the previous results, all crRNAs were designed to target the promoter adjacent regions of either template strands or coding strands. Plasmids expressing these crRNAs (Supplementary Data 1) were transformed into DF50ΔEPSΔcas3.

To test the effect of CRISPRi-mediated downregulation on genes located on megaplasmids, the *phaEC* gene cluster on megaplasmid pHM300 was chosen as a target. This gene cluster contained *phaE* (HFX_5220) and *phaC* (HFX_5221) genes which encoded PHBV synthase[38]. PHBV synthase is a key enzyme catalyzing the polymerization of 3HV-CoA and 3HB-CoA into PHBV. PHBV accumulation was completely abolished in *H. mediterranei* when *phaEC* genes were knockout. Thus, it is conceivable that the repression of *phaEC* would lead to a reduction in the transcription level of *phaEC*, and a consequent decrease in PHBV production. Two crRNAs were designed and used to repress the *phaEC* gene cluster, one binding to the template strand (phaEC-t1) of *phaE* (close to the promoter) and

one targeting the coding strand (phaEC-c1) in promoter region (Fig. 4a). An efficient downregulation of *phaEC* gene cluster was observed from both RT-qPCR analysis (Fig. 4b) and PHBV determination (Fig. 4c). The transcript level of *phaE* was reduced down to 43% by phaEC-t1, and 48% by phaEC-c1. Meanwhile, the transcript level of *phaC* was reduced down to 34% by phaEC-t1, and 29% by phaEC-c1. Thus, an obvious reduction in the mRNA levels of *phaE* and *phaC* was achieved by expressing phaEC-t1 or phaEC-c1. Similarly, PHBV production was reduced by expressing either of the two crRNAs. Consistently, after 8 days of cultivation, PHBV produced by the strain expressing phaEC-t1 and phaEC-c1 decreased by 39% and 15%, respectively. Taken together, the genes located on megaplasmids can be efficiently downregulated by the native CRISPRi system developed in *H. mediterranei*. Targeting the promoter adjacent region of a gene cluster resulted in repression of all the genes present in the cluster.

To test the effect of CRISPRi-mediated knockdown of essential genes, the genes encoding citrate synthase were targeted. Our bioinformatic analysis showed that HFX_0432 (*citZ*) and HFX_6170 (*gltA*) were two candidate genes encoding citrate synthase. Four crRNAs were designed to target the promoter region of *citZ* (Fig. 4d). Among them, three crRNAs (citZ-t1, t2, and t3) targeted the template strand and one crRNA (citZ-c1) targeted the coding strand. The RT-qPCR analysis showed that citZ-t2, t3, and c1 resulted in a strong and similar repression of the expression level ranging from 2 to 7%, compared to the control group (Fig. 4e). citZ-t1 reduced the expression level of *citZ* down to 75%. In the case of *gltA*, two crRNAs targeted the promoter region (gltA-t1) and the promoter adjacent region (gltA-t2) of the template strand, respectively, and one crRNA (gltA-c1) targeted the promoter region of the coding strand (Fig. 4d). The expression level of *gltA* was reduced down to 5% (gltA-t1), 56% (t2), or 50% (c1). (Fig. 4e). Among the three crRNAs, gltA-t1 exhibited the strongest repression effect on *gltA* expression. These results demonstrated that essential genes could be repressed efficiently by the CRISPRi system in *H. mediterranei*.

*H. mediterranei* possesses six CRISPR arrays, namely, C2, C18, C22, C26, P12, and P23[21]. All of them are actively transcribed and processed to generate mature crRNAs. Possibly, the existing

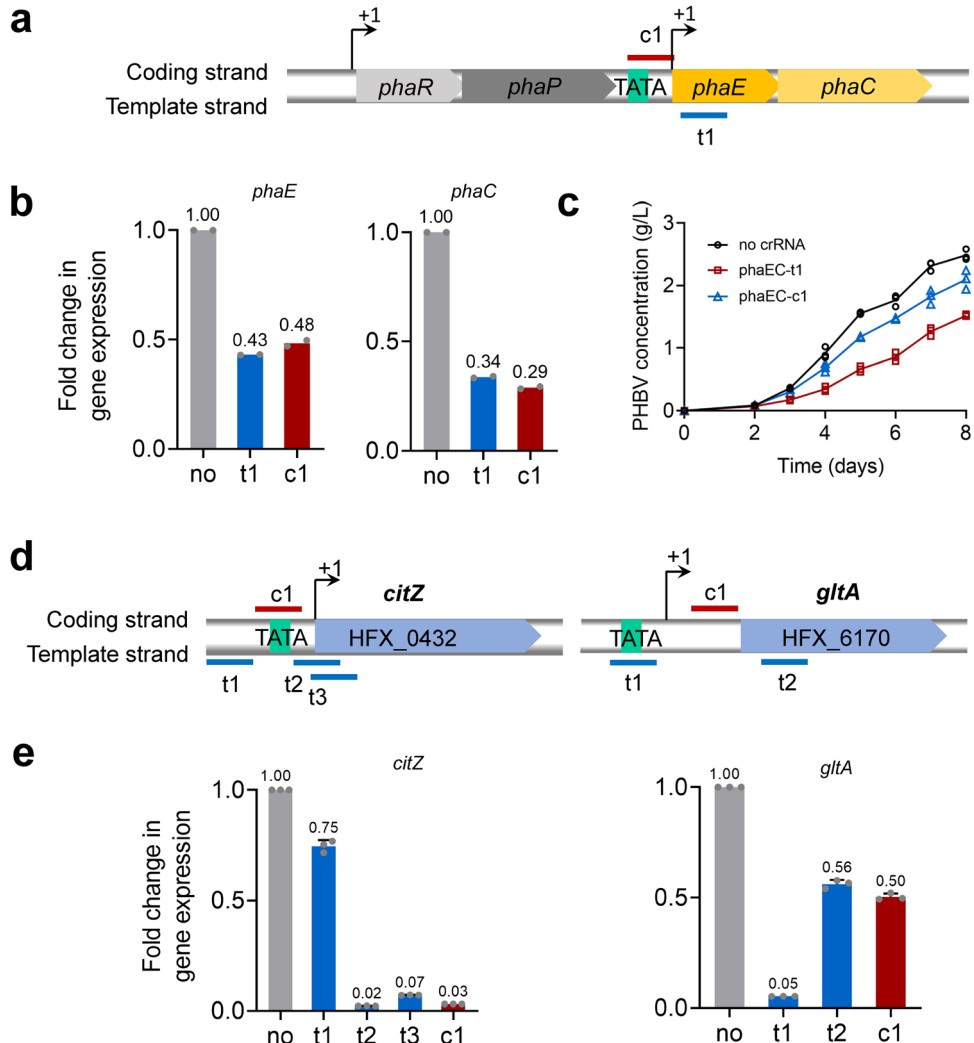

**Fig. 4 Repression of *phaEC* and citrate synthase genes using CRISPRi. a** Location of crRNA target sites in *phaEC*. *phaEC* are transcribed driven by two promoters of P*phaR* and a weaker promoter upstream of *phaEC*. t1 and c1 are designed to target the template strand and coding strand of *phaEC*, respectively. **b** RT-qPCR analysis of *phaE* and *phaC* expression levels. **c** PHBV production. **d** Location of crRNA target sites in *citZ* and *gltA*. *citZ* is transcribed from the A of the start codon ATG. t1, t2 and t3, and c1 are crRNAs targeting the template strand and coding strand of *citZ*, respectively. The TSS of *gltA* is 155 bases from the start codon. t1 and t2, and c1 are crRNAs targeting the template strand and coding strand of *gltA*, respectively. **e** Fold change in the expression of *citZ* and *gltA*. Data shown for two (**b**) or three (**c**, **e**) biological replicates. Error bars indicate SDs, *n* = 3.

CRISPR structure in *H. mediterranei* competed with the synthetic mini-CRISPRs for the Cas proteins. Thus, we speculated that the repression effect might be further enhanced by knockout of all six CRISPR loci in this strain. Three plasmids expressing different crRNAs (citZ-t1, citZ-t2, and phaEC-c1) were transformed into DF50ΔEPSΔ*cas3* and CRISPR-free (CRF) strain CRFΔEPSΔ*cas3*, respectively. The results showed that there was no substantial difference in the crRNA-guided repression between DF50ΔEPSΔ-*cas3* and CRFΔEPSΔ*cas3* (Supplementary Fig. 1). This indicated that the amount of native Cas proteins was sufficient for the CRISPRi system in *H. mediterranei*. Taken together, our native CRISPRi system repressed different types of genes effectively, including genes on the chromosome or megaplasmids, a single gene or a gene cluster, non-essential genes or essential genes, in *H. mediterranei*.

**CRISPRi repressed two genes/gene clusters simultaneously.** As the downregulation of a single gene worked well, we next tested the ability of our CRISPRi tool to downregulate two genes/gene clusters simultaneously in *H. mediterranei*. We selected the β-

ketothiolase-encoding genes *phaA* and *bktB* as targets. PhaA enzyme, involved in supplying 3-hydroxybutyl-CoA (3HB-CoA) during PHBV biosynthesis in *H. mediterranei*, was encoded by the cotranscribed HFX_1023 and HFX_1022 genes. Likewise, BktB enzyme, involved in supply of 3HB-CoA and 3-hydroxyvaleryl-CoA (3HV-CoA), was encoded by cotranscribed HFX_6004 and HFX_6003 genes[39]. One crRNA (phaA-t1) targeting the template strand of *phaA* and two crRNAs (bktB-c1 and c2) targeting the coding strand of *bktB* were designed and expressed in DF50ΔEPSΔ*cas3* (Fig. 5a). Individual expression of phaA-t1 and bktB-c2 reduced the respective transcription level of *phaA* and *bktB* down to 6% and 15% (Fig. 5b). On the other hand, bktB-c1 expression showed no gene silencing effect due to its distant upstream location of the promoter. Since, the two crRNAs phaA-t1 and bktB-c2 demonstrated an effective gene silencing, they were coexpressed to target *phaA* and *bktB*, simultaneously. Coexpression of the two crRNAs (DF50ΔEPSΔ*cas3*-AB) knocked down both genes, resulting in a severe reduction of transcription levels of *phaA* and *bktB* down to 5% and 3%, respectively (Fig. 5b).

In addition, the effect of *phaA* and/or *bktB* downregulation on PHBV accumulation by the respective strains was also evaluated

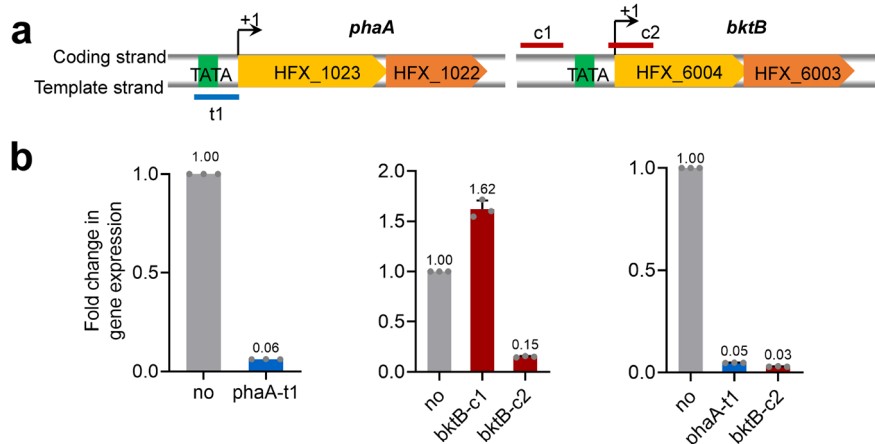

**Fig. 5 Repression of *phaA* and *bktB* individually and simultaneously. a** Location of crRNAs target sites in the *phaA* and *bktB* genes. They are co-transcribed from the A of the start codon ATG. t1 targets the template strand of *phaA* in promoter region. c1 and c2 targets the coding strand of *bktB*. **b** Changes in the transcriptional levels of *phaA* and *bktB* in DF50ΔEPSΔcas3 individually expressing phaA-t1, bktB-c1and, bktB-c1 or simultaneously expressing *phaA*-t1and *bktB*-c2. Data shown for three biological replicates. Error bars indicate SDs, $n = 3$.

### Table 1 PHBV accumulation by *H. mediterranei* strains with β-ketothiolases genes repressed via CRISPRi[a].

| Strains | CDW (g/L)[a] | PHBV content (%)[b] | PHBV concentration (g/L) | 3HV fraction (mol%) |
|---|---|---|---|---|
| DF50ΔEPSΔcas3-no crRNA | 4.59 ± 0.07 | 50.47 ± 1.56 | 2.32 ± 0.10 | 10.27 ± 0.81 |
| DF50ΔEPSΔcas3-phaA-t1 | 4.58 ± 0.23 | 44.19 ± 0.13 | 2.02 ± 0.11 | 10.81 ± 0.07 |
| DF50ΔEPSΔcas3-bktB-c2 | 4.93 ± 0.12 | 44.12 ± 1.88 | 2.17 ± 0.10 | 2.93 ± 0.47 |
| DF50ΔEPSΔcas3-AB | 2.31 ± 0.29 | 25.07 ± 3.77 | 0.58 ± 0.10 | 7.68 ± 0.59 |

All data are expressed as mean ± standard deviations from three independent experiments.
[a]CDW, dry weight of the cell (in grams) produced per liter of culture.
[b]PHBV content, the weight percent of PHBV in CDW.

(Table 1). It showed that after 8 days of fermentation in MG medium, the PHBV concentration of the strain expressing phaA-t1 alone decreased by 12.9% (from 2.32 to 2.02 g/L), with no obvious change in its 3HV molar fraction. Contrarily, the PHBV concentration of the strain expressing bktB-c2 alone did not change considerably, but its 3HV molar fraction was largely reduced by 71.5% (from 10.27 to 2.93 mol%). Interestingly, both the PHBV production and its 3HV molar fraction was greatly affected in DF50ΔEPSΔcas3-AB with both *phaA* and *bktB* inhibited simultaneously. The PHBV production decreased by 75% (from 2.32 to 0.58 g/L), and the 3HV molar fraction decreased by 25.2% (from 10.27 to 7.68 mol%). These results suggested that CRISPRi could orchestrate the simultaneous repression of two genes/gene clusters in *H. mediterranei*. In addition, the inhibition of *bktB* by CRISPRi changed the 3HV molar fraction of PHBV, indicating the biotechnological potential of CRISPRi in production of PHBV with desirable monomer incorporation from unrelated cheap carbon sources.

**CRISPRi repressed genes continuously.** In the previous sections, the CRISPRi-mediated repression was evaluated by determining the transcription level of all the target genes at a single time point (day 3 of cultivation). However, the inhibitory effect on the target genes throughout the cell growth period was not clear. Thus, to monitor the repression on the target genes during various growth phases of *H. mediterranei* in MG medium, two crRNAs citZ-t2 and gltA-t1 were expressed in DF50ΔEPSΔcas3 respectively. The transcription levels of *citZ* and *gltA* were evaluated every one day starting at day 2 of cultivation. From day 2 to day 7, the transcription level of *citZ* and *gltA* was reduced to 1.0–13.9% and 2.6–10.9%, respectively (Fig. 6a). Interestingly, the

repression after 2 days of cultivation was stronger than that at day 2. According to our previous study on the crRNA biogenesis of *H. mediterranei*, a limited number of mature crRNAs were generated in early exponential phase and the number gradually increased with cultivation time[24]. Thus, it was possible that more mature crRNAs were available for targeting during mid-exponential, and late-exponential phase, and stationary phase, resulting in a better gene repression than early exponential phase. The results showed that our CRISPRi system generated sustained and effective inhibition during the growth period of *H. mediterranei*.

**Repression of citrate synthase genes by plasmid-based CRISPRi enhanced PHBV synthesis.** Citrate synthase catalyzes the formation of citric acid from acetyl-CoA and oxaloacetate, which is the rate-limiting step in the TCA cycle. Acetyl-CoA, the substrate of citrate synthase, is also one of the important substrates for PHBV synthesis in *H. mediterranei*. Therefore, the reaction catalyzed by citrate synthase competes with the PHBV biosynthesis pathway for acetyl-CoA. It was speculated that down-regulation of the citrate synthase genes might decrease acetyl-CoA consumption by the TCA cycle, so that more acetyl-CoA are available for PHBV synthesis.

Our results found that the two crRNAs of citZ-t2 and gltA-t1 reduced the target gene (*citZ* and *gltA*) expression level down to 2% and 5%, respectively (Fig. 4e). Here, we expressed citZ-t2 and gltA-t1 individually as well as simultaneously in DF50ΔEPSΔcas3. The transcription of *citZ* or *gltA* was analyzed by RT-qPCR. Compared to the control strain containing no crRNA, *citZ* and *gltA* genes were repressed simultaneously with their expression levels reduced down to 7% and 21%, respectively in the strain CS

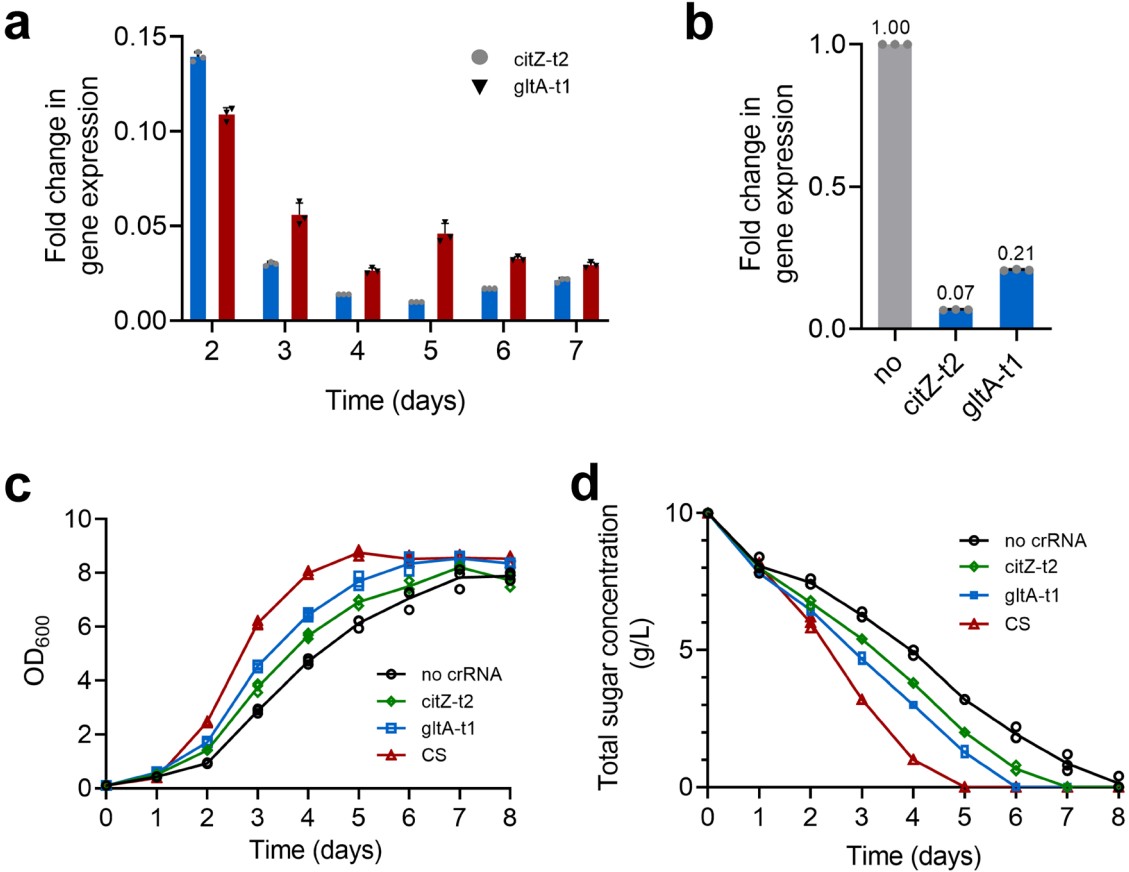

**Fig. 6 Continuous inhibition of *citZ* and *gltA* and the cell growth, glucose consumption of strains with these two genes repressed. a** The repression of *citZ* by citZ-t2 and *gltA* by gltA-t1 during the strain cultivation. **b** Changes in the transcriptional levels of *citZ* and *gltA* in DF50ΔEPSΔ*cas3* simultaneously expressing citZ-t2 and gltA-t1. **c** The cell growth of DF50ΔEPSΔ*cas3* with citrate synthase genes repressed. **d** Glucose consumption of DF50ΔEPSΔ*cas3* with citrate synthase genes repressed. CS represents DF50ΔEPSΔ*cas3* expressing both citZ-t2 and gltA-t1. Data shown for three biological replicates. Error bars indicate SDs, *n* = 3.

**Table 2 PHBV accumulation by *H. mediterranei* strains with citrate synthase genes repressed via plasmid-based CRISPRi.**

| Strains | CDW (g/L)[a] | PHBV content (%)[b] | PHBV concentration (g/L) | 3HV fraction (mol%) | PHBV productivity (g/L d) |
|---|---|---|---|---|---|
| *Plasmid-based crRNA expression system, 5 days cultivation* | | | | | |
| DF50ΔEPSΔ*cas3*-no crRNA | 3.63 ± 0.12 | 48.92 ± 1.15 | 1.78 ± 0.05 | 10.57 ± 0.29 | 0.36 ± 0.01 |
| DF50ΔEPSΔ*cas3*-citZ-t2 | 4.38 ± 0.09 | 58.72 ± 0.59 | 2.57 ± 0.04 | 10.10 ± 0.13 | 0.51 ± 0.01 |
| DF50ΔEPSΔ*cas3*-gltA-t1 | 4.69 ± 0.65 | 53.32 ± 0.46 | 2.50 ± 0.13 | 10.34 ± 0.55 | 0.50 ± 0.03 |
| DF50ΔEPSΔ*cas3*-CS | 5.66 ± 0.08 | 55.49 ± 1.75 | 3.14 ± 0.12 | 6.07 ± 0.21 | 0.63 ± 0.02 |
| *Chromosomal integration crRNA expression system, 2 days cultivation* | | | | | |
| DF50ΔEPSΔ*cas3*::*pyrF* | 8.29 ± 0.61 | 35.64 ± 0.19 | 2.88 ± 0.13 | 10.97 ± 0.28 | 1.44 ± 0.07 |
| DF50ΔEPSΔ*cas3*::*pyrF*::NT | 7.72 ± 0.16 | 36.47 ± 1.22 | 2.81 ± 0.04 | 10.83 ± 0.17 | 1.41 ± 0.02 |
| DF50ΔEPSΔ*cas3*::*pyrF*::CS | 9.07 ± 0.50 | 36.72 ± 1.18 | 3.33 ± 0.11 | 10.28 ± 0.44 | 1.67 ± 0.06 |

All data are expressed as mean ± standard deviations from three independent experiments and strains were cultivated in MG medium for 5 or 2 days.
[a]CDW, dry weight of the cell (in grams) produced per liter of culture.
[b]PHBV content, the weight percent of PHBV in CDW.

(Fig. 6b). All strains showed higher biomass and faster glucose consumption rate compared to the control (Fig. 6c, d). Especially, the strain CS resulted in the highest cell dry weight of 5.66 g/L and consumed the entire glucose at day 5 of cultivation. The PHBV accumulation was analyzed at day 5. All the three strains expressing crRNA exhibited a higher PHBV accumulation (Table 2). Notably, the strain CS exhibited 76.4% improvement in PHBV concentration (from 1.78 to 3.14 g/L) although the PHBV productivity was low (0.63 g/L d). Likewise, the strains expressing only one crRNA (citZ-t2 or gltA-t1) showed an increased PHBV production by 44.4% (from 1.78 to 2.57 g/L) or 40.4% (from 1.78 to 2.50 g/L). Nevertheless, the 3HV molar fraction of PHBV accumulated by the strain CS was the lowest (6.07 mol%). Probably, inhibition of citrate synthase genes redirected more acetyl-CoA towards 3HB monomer synthesis. However, the amount of 3HV monomer supplying was not increased, because its synthesis was limited by the concentration of its precursor propionyl-CoA. Therefore, when the synthesis of 3HB monomer increased, the 3HV molar fraction of PHBV decreased.

**Chromosomal integration of crRNA expression cassette facilitate PHBV synthesis**. To maintain the plasmid stability in plasmid-based crRNA expression system, fermentation medium devoid of yeast exact or uracil was used. Subsequently, the strains grew slowly and the PHBV productivity achieved was low. For the best strain CS, after 5 days of cultivation the final CDW was 5.66 g/L and PHBV productivity was only 0.63 g/L d (Table 2). In order to reduce the metabolic burden of plasmid, the CS expression cassette along with *pyrF* gene was integrated into the native position of *pyrF* in the genome of DF50ΔEPSΔ*cas3* (Supplementary Fig. 2). Two control stains, with chromosomal integration of *pyrF* or *pyrF* plus NT expression cassette targeting no sequence, were simultaneously constructed. The resulting strains were uracil auxotrophic and were grown in yeast exact supplemented MG medium. As expected, chromosomal expression system shortened the cultivation time from 5 to 2 days. In comparison with plasmid-based expression system, the inhibition of citrate synthase genes via chromosomal crRNA expression system led to an increase of 60.2% in CDW (from 5.66 to 9.07 g/L) and 165.1% in PHA productivity (from 0.63 to 1.67 g/L d), respectively (Table 2). Consistent with plasmid-based crRNA expression system, DF50ΔEPSΔ*cas3::pyrF::*CS exhibited faster cell grow and glucose consumption compared to the two controls (Supplementary Fig. 3). At day 2 of cultivation, chromosomal integration CS strain showed the highest CDW, PHBV concentration, and PHBV productivity among the three tested strains. It exhibited an increase of 17.5% in CDW (from 7.72 to 9.07 g/L), and 15.6% in PHA concentration/productivity (from 2.81 to 3.33 g/L; from 1.41 to 1.67 g/L d), respectively, compared to the control NT. Therefore, chromosomally integrated crRNA expression system was more preferable over the plasmid-based expression system for improving PHBV synthesis in *H. mediterranei* from industrial perspective.

**Repression of citrate synthase expression redirected carbon metabolic flux**. We have tried to redirect metabolic flux towards PHBV synthesis by downregulating citrate synthase genes using CRISPRi in *H. mediterranei*. Because the strain CS exhibited the highest PHBV production, this strain was chosen for transcriptome analysis. The transcriptome sequencing and analysis showed that there were 1427 genes whose transcription levels changed significantly ($P < 0.05$, |log$_2$ (fold change)| ≥1). Among them, 759 genes were upregulated and 668 genes were downregulated. Significant changes in the expression of 1427 genes indicated that the inhibition of citrate synthase genes had a major impact on metabolic network of cells, involving multiple biological processes such as TCA cycle, glyoxylate cycle, pyruvate metabolism, amino acids metabolism, fatty acid metabolism, and biosynthesis of secondary metabolites.

The differential expression analysis showed that *citZ* and *gltA* were significantly downregulated (Fig. 7), which was consistent with the results of RT-qPCR (Fig. 6b). Besides, the genes encoding several other enzymes involved in TCA cycle were also downregulated to different levels (Fig. 7), such as the isocitrate dehydrogenase gene, the α-ketoglutarate hydrogenase genes, the succinyl-CoA synthetase genes and the fumarase genes. This indicated that the TCA cycle was repressed after the inhibition of the citrate synthase genes. In addition, it was observed that HFX_2079 encoding the key enzyme of the glyoxylate cycle was significantly upregulated. Moreover, the expression of phosphoenolpyruvate synthase (PPS) and phosphoenolpyruvate (PEP) carboxylase, which catalyzed pyruvate to oxaloacetate conversion via PEP formation was significantly upregulated. It was speculated that upregulation of the glyoxylate cycle and pyruvate to oxaloacetate conversion facilitated and compensated the

formation of 4-carbon compounds, when the TCA cycle was suppressed in *H. mediterranei*.

The differential expression of genes related to PHBV metabolism was also analyzed. Among two acetoacetyl-CoA reductases (PhaB1 and PhaB2), PhaB2 plays the major role in monomer supply[40]. Interestingly, *phaB2* expression was significantly upregulated in the CS strain, which implied that monomer supplying pathway was enhanced. Furthermore, the expression of genes related to PHBV synthesis (*phaE*, *phaC*, and *phaC3*) and PHBV regulation (*phaR* and *phaP*) was significantly upregulated. These findings indicated that repression of citrate synthase genes reduced the supply of acetyl-CoA towards TCA cycle, leading to downregulation of the latter. As a consequence, more acetyl-CoA molecules were converted to 3HB-CoA and 3HV-CoA monomers, leading to the upregulation of PHBV monomer supply pathway and PHBV synthesis. All these factors finally promoted the PHBV accumulation in the CS strain.

## Discussion

In our present study, totally fifteen PAM sequences were tested by employing plasmid-based invader assay in DF50ΔEPS. TTG and TTC were two efficient functional PAMs for *H. mediterranei* as they exhibited a considerably lower number of transformants in the plate. In *H. hispanica*, four functional PAMs, TTC, TTG, TTT, and CCC, were separately recognized by its endogenous I-B system to elicit CRISPR interference[35]. Similarly, the trinucleotide sequences ACT, TAA, TAT, TAG, TTC, and CAC served as functional PAMs in *H. volcanii*[36]. All the three haloarchaea, *H. mediterranei*, *H. hispanica*, and *H. volcanii* possessed the same type of endogenous CRISPR-Cas system (type I-B) and nearly identical repeat sequences. However, the functional PAMs during interference in these three haloarchaea were not very conserved. This indicated that the PAM sequences for interference are somewhat host specific. Notably, their Cas proteins are less conserved, suggesting that the PAM selectivity might depended on the different Cas proteins recruited by the different host organisms.

CRISPRi is an RNA-guided tool where crRNA is one of the key factors responsible for efficient gene regulation. The CRISPRi tool based on the endogenous type I-B CRISPR-Cas system of *H. volcanii* was the first CRISPRi technique developed in archaea[30]. In this system, the inhibitory effect of the CRISPRi depended on the target region of the crRNA. Usually, the optimal crRNA binding site was localized on the template strand, between −70 to +20 from TSS[31]. Moreover, the crRNAs targeting near TSS led to a more effective gene silencing than that targeting the open reading frame in *H. volcanii*. It was also believed that the crRNAs targeting the template strand led to much more effective inhibitory effects than those targeting the coding strand in this strain. Interestingly in *H. mediterranei*, crRNAs targeting sites near the TSS on either the template strand or the coding strand generated an effective inhibition, a phenomenon which is different from the strand bias observed in *H. volcanii*. This feature makes the design of crRNAs for *H. mediterranei* more flexible. Moreover, unlike *H. volcanii*, crRNAs targeting sites down far from the TSS are still effective in *H. mediterranei*. Such differences might be due to the higher crRNA amount expressed by the strong constitutive P$_{phaR}$ promoter used in *H. mediterranei*.

Citrate synthase is an essential enzyme that utilizes acetyl-CoA as its substrate to generate citric acid in TCA cycle. Since, acetyl-CoA is also a substrate for PHBV monomer supplying pathway, the citrate synthase gene has been targeted in several microbial strains to achieve a higher PHA production. *E. coli* does not produce PHA naturally as it lacks PHA biosynthetic pathway. Repression of *gltA* gene encoding citrate synthase using CRISPRi

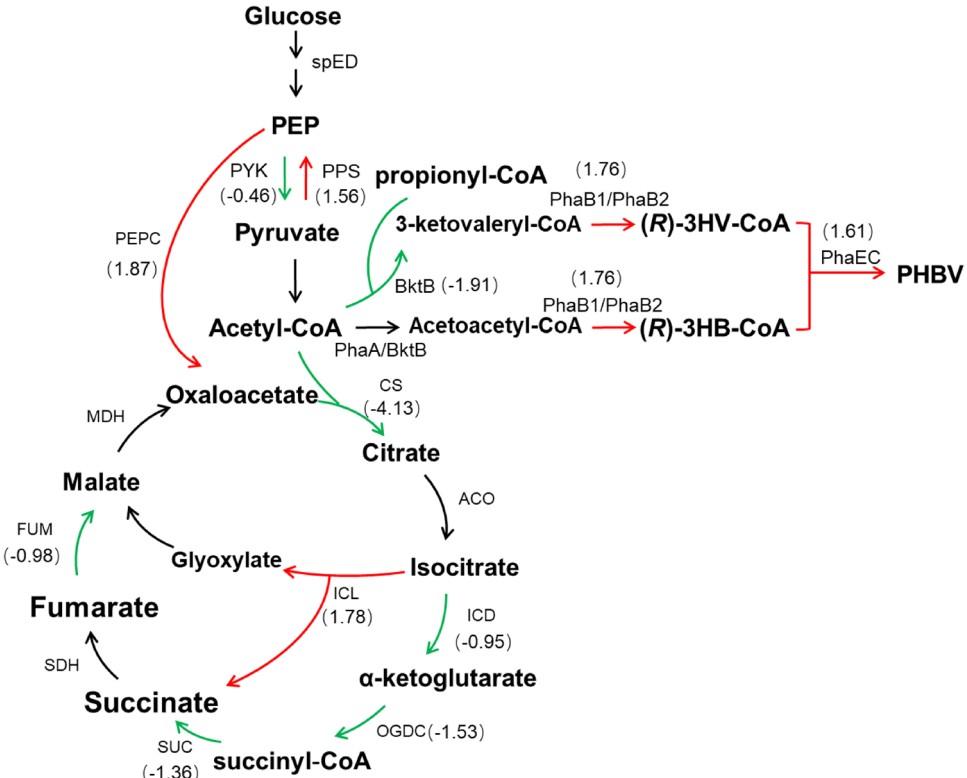

**Fig. 7 Effects of inhibition of citrate synthase genes on metabolism of *H. mediterranei*.** Red and green arrows indicate upregulation and downregulation of gene expression, respectively. Numbers in brackets are log$_2$ (fold change). CS citrate synthase, ACO aconitase, ICD isocitrate dehydrogenase, OGDC α-ketoglutarate dehydrogenase, SUC succinyl-CoA synthetase, SDH succinate dehydrogenase, FUM fumarase, MDH malate dehydrogenase, PEP phosphoenolpyruvate, PEPC, PEP carboxylase, PYK pyruvate kinase, PPS phosphoenolpyruvate synthase.

slowed down its cell growth, glycerol consumption, and increased acetate accumulation. However, introduction of the PHB synthetic pathway redirected acetyl-CoA towards PHB synthesis, leading to a 3.4-fold increase in PHB production[32]. Similarly, controlled repression of *gltA* gene by CRISPRi improved PHB production by 8% in *Halomonas* sp. TD01[34]. Although the strain experienced a prolonged lag growth phase, the final cell density was similar to the control strain. As expected, repression of the two citrate synthase genes in *H. mediterranei* by CRISPRi increased the PHBV content in the engineered strain by 76.4%, compared to the control strain. Moreover, the cell growth and glucose consumption were faster and no noticeable accumulation of acetate was observed in the engineered strain.

Plasmid-based system employed in this study to express the crRNA module is a conventional method of gene expression. However, plasmid instability is a primary problem in this approach[41]. Maintenance of plasmid requires additional selective pressure like use of selective media or addition of antibiotics[42]. However, such types of selection strategies incur additional cost and even imposes metabolic burden on the host cell. In our case, fermentation medium without yeast extract or uracil was used to maintain plasmid stability. It retarded the cell growth which prolonged the cultivation time to 5 days. To address this issue, the crRNA was expressed chromosomally without need of any selective media. Chromosomal integration provided increased stability and circumvented the problem of retarded cell growth. The engineered strain reached an even higher PHA production level than that of plasmid-based system within 2 days of cultivation. After 3 days of fermentation, PHA concentration was further increased to 4.13 g/L, respectively (Supplementary Table 1). With increasing time, glucose got exhausted that might have led to some slight increase in cell growth (CDW of 9.16 g/L),

and an even decrease in PHA productivity (1.38 g/L d) at day 3 of cultivation. Hence, the feeding strategy will be optimized to further improve cell growth and PHA production in the CRISPRi-engineered *H. mediterranei* in our following research. Taken together, chromosomal integration is a promising technique that not only increases stability of gene expression but also may realize large-scale production in short time.

CRISPRi is also a versatile genetic tool that can be potentially applied to control monomer composition of PHA. *Halomonas* sp. TD01 produces PHBV when propionate was supplemented with glucose. The former is converted to 3HV precursor molecule, propionyl-CoA, which is further converted to 2-methylcitrate, by 2-methylcitrate synthase (encoded by *prpC*), and enters the methyl citric acid cycle. Interestingly, repression of *prpC* redirected more propionyl-CoA towards PHBV synthesis and improved 3HV molar fraction up to 13% in this strain by using CRISPRi technology[34]. In contrast, the 3HV molar fraction of PHBV was reduced by 71.5% due to less 3HV monomer generated when the expression of BktB was repressed by CRISPRi tool in *H. mediterranei*. The inhibition of two citrate synthase genes via plasmid-based CRISPRi tool redirected more acetyl-CoA towards 3HB monomer supplying, and thus reduced 3HV molar fraction. With the successful application of CRISPRi in archaeal metabolic engineering, as demonstrated in this study, it is further expected that this tool might be even implemented to enhance 3HV monomer incorporation in PHBV. In another interesting study, controlled biosynthesis of poly(3-hydroxybutyrate-*co*-4-hydroxybutyrate) [P(3HB-*co*-4HB)] was achieved using CRISPRi in *E. coli*[43]. Therefore, CRISPRi is a powerful tool that can be employed not only to improve PHA production but also to realize synthesis of PHA with tuneable monomer ratio.

## Methods

**Strains, medium, and culture conditions**. All the strains used in this study are listed in Supplementary Table 2. *E. coli* JM109 was used for plasmid construction and *E. coli* JM110 was used to eliminate the methylated plasmids in vivo. *E. coli* was cultivated in LB medium at 37 °C[44]. When needed, ampicillin was added to a final concentration of 100 mg/L. The uracil auxotrophic strain *H. mediterranei* DF50ΔEPS (*pyrF* and EPS gene cluster deleted) and its derivative strains were cultivated in AS-168 medium[45] with uracil added at a concentration of 50 mg/L. When needed, 5-fluoroorotic acid (5-FOA) was added to a final concentration of 250 mg/L. *H. mediterranei* strains containing plasmids were grown in AS-168SY medium (AS-168 medium with yeast extract subtracted). MG medium[5] was used for lycopene production and PHBV production. For lycopene production, 50BΔ2549Δ*cas3* carrying plasmids were inoculated into MG medium at an initial OD$_{600}$ of 0.1 and cultivated at 37 °C, 200 rpm for 3 days. For PHBV production, seed cultures of DF50ΔEPSΔ*cas3* carrying crRNA expression plasmids or crRNA expression cassette in chromosome were grown in MG medium for 48 h at 37 °C, 200 rpm. Then seed cultures were inoculated in 50 mL MG medium or PHA production medium in 250 mL shake flasks and cultured for 8 days or 3 days. The initial OD$_{600}$ of each culture was adjusted to 0.1.

**Plasmid-based invader assay to determine functional PAM sequences**. Different trinucleotide combinations designed as PAM sequences were introduced upstream of spacer1 of CRISPR locus P23 (P23-S1) of *H. mediterranei*. Sticky ends, PAM sequence and P23-S1 were synthesized as oligonucleotides. Two complementary oligonucleotides (Supplementary Data 2) were annealed to generate a sticky fragment. Then different sticky fragments were respectively inserted into pWL502[46] digested with *BamH* I and *Kpn* I, using One Step Cloning Kit (Yeasen, China). These pWL502-based invader plasmids (Supplementary Data 1) were shuttled to *E. coli* JM110 cells to eliminate plasmid methylation and then finally transformed into *H. mediterranei* DF50. Plasmid pWL502 was transformed as a positive control. Transformants were selected on AS-168SY plates without uracil. Transformation rates were calculated as the number of colonies obtained by transformation of 1 μg of plasmid DNA (cfu/μg of DNA). Interference effect was judged by relative transformation efficiency compared with the positive control. For each invader plasmid, two or three replicates were performed to evaluate the interference effect.

**Construction of gene knockout mutants**. Two important gene knockout mutants, DF50ΔEPSΔ*cas3* and 50BΔ2549Δ*cas3*, were obtained by deleting *cas3* gene in DF50ΔEPS and 50BΔ2549, respectively. Plasmids for gene knockout (Supplementary Data 1) were constructed based on suicide plasmid pHFX[15] or pHFX-B60. The primers used for gene knockout are summarized in Supplementary Data 2. Upstream and downstream fragments (~500 bp) of target genes were separately amplified and then linked together by bridge PCR method. These linked fragments (~1 kb) were then inserted into pHFX or pHFX-B60 digested with *BamH* I and *Kpn* I, using One Step Cloning Kit (Yeasen, Co., Ltd., China). The constructed knockout plasmids were validated by DNA sequencing and then transformed into DF50ΔEPS, and its derivative strains using the polyethylene glycol-mediated transformation method[47]. The mutants were screened and verified via PCR by using the primers listed in Supplementary Data 2[45].

**Plasmid-based or chromosome-based crRNA expression**. The plasmids carrying a mini-CRISPR array were constructed to express crRNAs. The mini-CRISPR array contained a short-version P$_{phaR}$ promoter[48], a specific spacer flanked by two identical repeats and a T$_8$ terminator and synthesized by GenScript company (Nanjing, China) (Supplementary Data 2). The spacer was selected specifically for each target as a 35 nt sequence, which must match to the protospacer downstream of a PAM in the target. The synthetic mini-CRISPR arrays were amplified with primers crRNA-F/R and inserted into pWL502, generating pWL502-crRNA (Supplementary Data 1). After validation by DNA sequencing, pWL502-crRNA was transformed into 50BΔ2549Δ*cas3* or DF50ΔEPSΔ*cas3* to express crRNAs against target genes.

For chromosomal integration of crRNA expression cassette and *pyrF* complementation (Supplementary Fig. 2), pMD18-T-*pyrF*, pBM23-*pyrF*-NT, and pBM23-*pyrF*-CS were first constructed in *E. coli* JM109 (Supplementary Data 1). The *pyrF* gene with its upstream and downstream fragments were amplified with primers *pyrF*-1/4 and inserted into pMD18-T, generating pMD18-T-*pyrF*. The *pyrF* with its upstream and downstream fragments and the mini-CRISPR array (CS/NT) were amplified with primers *pyrF*-1/2, CS-F/R and *pyrF*-3/4 and then inserted into pBM23, generating pBM23-*pyrF*-NT and pBM23-*pyrF*-CS, respectively. After validation by DNA sequencing, the plasmids were used as templates to clone the integrated fragments. The linear integrated fragments were transformed into DF50ΔEPSΔ*cas3* for homologous integration into the chromosome. The integrated strains were screened on AS-168SY plates and validated by primers *pyrF*-1/4. All of the primers used are listed in Supplementary Data 2.

**Lycopene quantification**. The liquid cultures (1 mL) of *H. mediterranei* 50BΔ2549Δ*cas3* containing crRNA expression plasmids were centrifuged at 12,000×*g*, 4 °C, for 5 min. Then the pellet was resuspended in 1 mL of acetone under a reduced light condition to prevent photobleaching and degradation. The acetone supernatant containing lycopene was collected by centrifugation at 12,000 × *g*, 4 °C, for 3 min, and transferred to a new tube. This process was repeated until the pellets were completely white. The same volume of acetone was used for the lycopene extraction of each sample. The UV–Vis absorbance spectrum of the acetone supernatant containing lycopene was measured using a N5000 spectrophotometer (Yoke Instrument, China). The wavelength range was set as 350–600 nm. The fold change in lycopene production was calculated as the absorbance at 471 nm per unit OD$_{600}$ of the culture, compared with the control without mini-CRISPR.

**RNA extraction and RT-qPCR analysis**. *H. mediterranei* cells were cultured in MG medium at 37 °C. The total RNA was isolated from exponentially growing cells using TRIzol reagent (Invitrogen, USA)[49]. RT-qPCR was used to analyze the gene transcriptional level. Ten microgram of total RNA was digested with TURBO DNA-free™ Kit (Thermo Fischer Scientific, USA) to remove DNA from samples. Then cDNA was synthesized from the DNA-free RNA samples using random hexamer primers and Moloney Murine Leukemia Virus Reverse Transcriptase (MMLV-RT) (Promega, USA). All real-time PCR were performed using KAPA™ SYBR® Fast qPCR Kit (KAPA Biosystems, USA). The fold change of gene expression was analyzed by ViiA™ 7 Real-Time PCR System (Applied Biosystems, Inc., USA). 7S rRNA was used as the inner standard. The primers used are listed in Supplementary Data 2. RT-qPCR were performed in two or three replicates.

**PHBV accumulation analysis**. Ten to thirty milliliter of cultures were collected by centrifugation at 10,000 × *g* for 20 min. Then the pellets were lyophilized and weighed. The lyophilized cells were treated with 2 mL chloroform and 2 mL methanol containing 3% (v/v) sulfuric acid and 1 g/L benzoic acid at 100 °C for 4 h. After cooling to room temperature, 1 mL deionized water was added. The samples were then mixed thoroughly and stood still for 3 h for stratification. The chloroform solution from the bottom layer was taken by syringe for gas chromatograph analysis (GC-6820, Agilent, USA)[50]. Benzoic acid was used as an internal standard. The PHBV concentration (g/L) was calculated as mass of PHBV/volume of cultures collected. For each experimental group, three parallel samples were set.

**Transcriptome analysis**. The total RNA was extracted from the cells cultured in MG medium for 3 days. After checking the RNA purity and integrity, a total amount of 3 μg RNA per sample was used to generate sequencing library using NEB Next® Ultra™ RNA Library Prep Kit for Illumina® (NEB, USA). After cluster generation, the library preparations were sequenced on an Illumina Novaseq platform and 150 bp paired-end reads were generated (Novogene Co., Ltd., Beijing, China). Clean data (clean reads) were obtained by removing reads containing adapter, reads containing ploy-N and low-quality reads from raw data. All the downstream analyses were based on the clean data with high quality. To estimate gene expression levels, HTSeq v0.6.1 was used to count the reads numbers mapped to each gene. And then FPKM (expected number of Fragments Per Kilobase of transcript sequence per Million base pairs sequenced) of each gene was calculated based on the length of the gene and reads count mapped to this gene. Three biological replicates were set for each group.

**Statistics and reproducibility**. Data was analyzed with GraphPad Prism™ and shown as means ± standard deviation (SD). Differential expression analysis of two groups was performed using the DESeq R package. The *p*-value of 0.05 and log$_2$ (Fold change) of 1 were set as the threshold for significantly differential expression.

**Reporting summary**. Further information on research design is available in the Nature Research Reporting Summary linked to this article.

## Data availability

The RNA-seq data has been deposited in the China National Microbiology Data Center (www.nmdc.cn) under accession number NMDC10017751. Plasmids have been deposited in Addgene under accession numbers 174382-174385, 174387-174390, and 174396. All other data are available from the corresponding author on reasonable request. Raw data underlying all figures have been provided as Supplementary Data 3.

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

## Acknowledgements

This work was supported by grants from Funding from National Key R&D Program of China (No. 2020YFA0906800), the National Natural Science Foundation of China (No. 31970031 and No. 91751201), and Strategic Priority Research Program of the Chinese Academy of Sciences (No. XDA24020101).

## Author contributions

L.L.: Methodology, investigation, formal analysis, data curation, writing—original draft. J.C.: Investigation and data curation. R.M.: Data curation and writing—review and editing. Q.G.: Investigation. F.C. and T.X.: Methodology. Z.Z.: Methodology and resources. H.X. and J.H.: Methodology, supervision and writing—review and editing.

## Competing interests

The authors declare no competing interests.
