## [Peer Review File · Communications Biology]

Reviewers' comments:

Reviewer #1 (Remarks to the Author):

The MS submitted by Lin Lin and coworkers summarizes a study in which CRISPRi technology is applied in a haloarchaea to redirect the flow of C so that the synthesis of PHBV biopolymers is prioritized. The topic is of interest while PHBV is a degradable bioplastic that can be an alternative to chemical synthetic plastic. It fits to the general scope of the journal and it is well organized.

However, the MS has serious flaws: concept errors in the introduction, lack of details in the methodology so that experiments can be reproducible, unclear experimentation approach (the synthesis of genes involved in the synthesis of lycopene is repressed and the loss of tonality makes them speculate that it is due to lower levels of lycopene; they do not quantify or identify the pigment. With this, carbon flux is assumed to be derived from carotenoid synthesis to PHBV synthesis). Several phrases such as the following one are used to justify the novelty of the work "Our results indicated that the CRISPRi system could be used to achieve different levels of gene regulation in *H. mediterranei*." However, the authors ignore that CRISPR-Cas was first described in *H. mediterranei*, so there is nothing new in which this technology works in this species.

On the other hand, error bars are missing in some plots and English must be revised by a native English speaker.

Comments have been embedded through the MS in order to help the authors.

Reviewer #2 (Remarks to the Author):

The paper entitled "CRISPRi-mediated carbon flux redirecting to improve poly(3-hydroxybutyrate-co-3-hydroxyvalerate) production in haloarchaea" provides a very important study ready for publication after minor revisions:

- 1) The term CDW should be defined in the legend of Table 1
- 2) The term CDW should be also defined in the legend of Table 2
- 3) The introduction section could be improved regarding to the increasing importance of PHBV in a wide range of industrial applications. Thus, for example, in Line 53 and 54, the sentence "Poly (3-hydroxybutyrate) (PHB) and PHBV are two extensively researched PHAs" could be improved indicating for example that the physical and biological properties of these biopolymers can be enhanced following a wide range of developed strategies for a great number of industrial applications (<https://doi.org/10.3390/polym10070732>; <https://doi.org/10.1016/j.ijbiomac.2019.10.034>)

Reviewer #3 (Remarks to the Author):

The manuscript under review presents the results of a CRISPR based approach to increasing biopolymer (PHBV) production in *H. mediterranei*. Overall the work is interesting, reasonably well written and sufficiently novel to warrant eventual publication. The work is within the scope of *Commsbio*. There are however, several points which need to be addressed, prior to publication.

- The standard of English language used throughout the paper is generally pretty good, though there is scope for some improvement. Another proof read will likely be sufficient.

- Abstract; needs to be re-written, to focus on the main findings of the work presented and their importance/impact. The generic description of CRISPRi should be removed. When reporting percentage increases the actual figures (e.g. PHBV concentrations) should be quoted.

- Introduction; what is the main motivation behind the work and what is the justification for the

approach adopted? There are lots of ways in which PHBV productivity can be increased, why a genetic manipulation approach? What are the advantages. At present the introduction basically says that there is an interesting technique that hasn't been applied to *H. mediterranei* before - more robust justification than this is required.

-Results; This section requires a general description of the approach used and an explanation of the logic applied, in order to help guide the reader through the work. In general the results seem to be reliable and of good quality. Data are presented clearly in figures and tables and there is clearly novelty. What is the link between cell growth and PHBV production in the engineered strains? Can the authors provide an argument that it isn't just the fact that there are more cells which leads to more polymer (this doesn't seem to be the case, but this does need to be commented on).

Discussion - Add a statement about how 3HV fraction in the co-polymer is influenced by the various genetic changes made. Though there are various comments on this throughout these separate strands need to be brought together.

- Methods; generally well described and appropriate.

Point-to-Point Response to Reviewers' Comments

(Manuscript ID# COMMSBIO-21-1274-T)

Reviewer #1

Comments to the Author:

The MS submitted by Lin Lin and coworkers summarizes a study in which CRISPRi technology is applied in a haloarchaea to redirect the flow of C so that the synthesis of PHBV biopolymers is prioritized. The topic is of interest while PHBV is a degradable bioplastic that can be an alternative to chemical synthetic plastic. It fits to the general scope of the journal and it is well organized.

Response: Great thanks for your positive comment and helpful suggestions.

1. The MS has serious flaws: concept errors in the introduction, lack of details in the methodology so that experiments can be reproducible, unclear experimentation approach (the synthesis of genes involved in the synthesis of lycopene is repressed and the loss of tonality makes them speculate that it is due to lower levels of lycopene; they do not quantify or identify the pigment. With this, carbon flux is assumed to be derived from carotenoid synthesis to PHBV synthesis).

Response: Thanks for your corrections. Some errors in the introduction have been corrected in the modified manuscript (lines 95 and 99). In the methodology, more details have been included to ensure reproducible experimental results (lines 520, 549-550, 559, 561, 569-578, 580-582, and 632.).

The confusion raised by the reviewer might be caused by our unclear description about the host strains used for gene repression. We have made it clear by including more detailed information (lines 297-298, 525-528 and 584). Based on our previous study about lycopene synthesis in *H. mediterranei* (Zuo et al. 2018, *Frontiers in Microbiol.* 9), the wild-type strain could not synthesize detectable lycopene and we achieved an engineered strain 50B Δ 2549 with high lycopene productivity. The gene of *crtI* encodes phytoene desaturase that catalyzes the formation of lycopene from phytoene. In the present study, when evaluating the gene repressive effect of crRNAs targeting different regions, *crtI* was used as the target gene in the engineered strain 50B Δ 2549 Δ *cas3* which exhibited high lycopene productivity. Thus, we could determine the gene inhibition based on cell color and lycopene quantification. When studying the repression effect of the CRISPRi system on different types of genes, a noncarotenoid-producing mutant named DF50 Δ EPS Δ *cas3* with EPS gene cluster and *cas3* gene deleted was used as the host strain. This strain was also used to repress the citrate synthase genes to enhance PHBV production. Our findings suggested that

metabolic flux from central metabolic pathways was redirected to PHBV synthesis pathway. It should be noted that we did not engineer the carotenoid biosynthetic pathways in DF50 Δ EPS Δ cas3 and our study did not suggest that carbon flux directed to PHBV synthesis was derived from carotenoid synthesis.

Reference:

Zuo, Z. Q. et al. Engineering *Haloferax mediterranei* as an efficient platform for high level production of lycopene. *Frontiers in Microbiol.* 9 (2018).

2. Several phrases such as the following one are used to justify the novelty of the work "Our results indicated that the CRISPRi system could be used to achieve different levels of gene regulation in *H. mediterranei*." However, the authors ignore that CRISPR-Cas was first described in *H. mediterranei*, so there is nothing new in which this technology works in this species.

Response: Thanks for your comment. The comment raised by the reviewer might be caused by our unclear description. In 1993, some sequences similar to CRISPR loci were found in *H. mediterranei* (Mojica, F. et al., 1993, *Mol. Microbiol.* 9, 613-621). Ten years later, our group reported the presence of subtype I-B CRISPR-Cas system in *H. mediterranei* (Li, M. et al. 2013, *J. Bacteriol.* 195, 867-875). However, function validation of the CRISPR-Cas system has not been reported yet in this strain. In the present study, we identify the functional PAM sequence which is a prerequisite for developing CRISPRi in *H. mediterranei*. We develop CRISPRi in *H. mediterranei* and apply it to metabolic engineering to improve PHBV biosynthesis. Our results demonstrate that the developed CRISPRi is an efficient tool for precise regulation of gene expression, and further fine-tune modulation of metabolic flux.

References:

1. Mojica, F. J. M., Juez, G. & Rodríguez-Valera, F. Transcription at different salinities of *Haloferax mediterranei* sequences adjacent to partially modified PstI sites. *Mol. Microbiol.* 9, 613-621 (1993).
2. Li, M. et al. Characterization of CRISPR RNA biogenesis and Cas6 cleavage-mediated inhibition of a provirus in the haloarchaeon *Haloferax mediterranei*. *J. Bacteriol.* 195, 867-875 (2013).

3. Error bars are missing in some plots and English must be revised by a native English speaker.

Response: Thanks for your comment. We have optimized the plot presentation to make the error bars much clearer (Fig. 6C and 6 D, and Supplementary Fig. 3). The whole manuscript has been revised and polished to improve its English level by native English speakers.

Reviewer #2

Comments to the Author:

The paper entitled “CRISPRi-mediated carbon flux redirecting to improve poly(3-hydroxybutyrate-co-3-hydroxyvalerate) production in haloarchaea” provides a very important study ready for publication after minor revisions.

Response: Great thanks for your positive comment and helpful suggestions.

1. The term CDW should be defined in the legend of Table 1.

Response: Thanks for your comment. The term CDW has been redefined as “dry weight of the cell (in grams) produced per liter of culture” in the footnotes of Table 1.

2. The term CDW should be also defined in the legend of Table 2.

Response: Thanks for your comment. The term CDW has been redefined as “CDW, dry weight of the cell (in grams) produced per liter of culture” in the footnotes of Table 2.

3. The introduction section could be improved regarding to the increasing importance of PHBV in a wide range of industrial applications. Thus, for example, in Line 53 and 54, the sentence “Poly (3-hydroxybutyrate) (PHB) and PHBV are two extensively researched PHAs” could be improved indicating for example that the physical and biological properties of these biopolymers can be enhanced following a wide range of developed strategies for a great number of industrial applications (<https://doi.org/10.3390/polym10070732>; <https://doi.org/10.1016/j.ijbiomac.2019.10.034>).

Response: Great thanks for the information. In the introduction section, we have added the increasing importance of PHBV in industrial applications and have included that the enhancement of physical and biological properties of these biopolymers will improve their applicability (lines 55-57, and 59-62). The two papers recommend by the reviewer have been cited.

References:

1. Rivera-Briso, A. L. & Serrano-Aroca, A. Poly(3-hydroxybutyrate-co-3-hydroxyvalerate): Enhancement strategies for advanced applications. *Polymers (Basel)*. 10 (7), 732 (2018).
2. Rivera-Briso, A. L., Aachmann, F. L., Moreno-Manzano, V. & Serrano-Aroca, A. Graphene oxide nanosheets versus carbon nanofibers: Enhancement of physical and biological properties of poly(3-hydroxybutyrate-co-3-hydroxyvalerate) films for biomedical applications. *Int. J. Biol. Macromol.* 143, 1000-1008 (2020).

Reviewer #3

Comments to the Author:

The manuscript under review presents the results of a CRISPR based approach to increasing biopolymer (PHBV) production in *H.mediterranei*. Overall the work is interesting, reasonably well written and sufficiently novel to warrant eventual publication. The work is within the scope of Commsbio. There are however, several points which need to be address, prior to publication.

Response: Great thanks for your positive comments and valuable suggestions. We have addressed all the comments one by one.

1. The standard of English language used throughout the paper is generally pretty good, though there is scope for some improvement. Another proof read will likely be sufficient.

Response: Thanks for your comment. The manuscript has been revised by native English speakers to further improve the English level. Another proof reading has been done.

2. Abstract; needs to be re-written, to focus on the main findings of the work presented and their importance/impact. The generic description of CRISPRi should be removed. When reporting percentage increases the actual figures (e.g. PHBV concentrations) should be quoted.

Response: Thanks for your comment. Based on your suggestion, we have rewritten the abstract (lines 21-37).

The haloarchaeon *Haloferax mediterranei* is a potential strain for poly(3-hydroxybutyrate-co-3-hydroxyvalerate) (PHBV) production, yet the production yield and cost are the major obstacles hindering the use of this archaeal strain. Leveraging the endogenous type I-B CRISPR-Cas system in *H. mediterranei*, we develop a CRISPR-based interference (CRISPRi) approach that allows to regulate the metabolic pathways related to PHBV synthesis, thereby enhancing PHBV production. Our CRISPRi approach can downregulate the gene expression in a range of 25% to 98% depending upon the target region. Importantly, plasmid-mediated CRISPRi downregulation on the citrate synthase genes (*citZ* and *glTA*) improves the PHBV accumulation by 76.4% (from 1.78 to 3.14 g/L). When crRNA cassette integrated into chromosome, this further shortens the PHBV fermentation period and enhances PHA productivity by 165%. Our transcriptome analysis shows that repression of citrate synthase genes redirects metabolic flux from the central metabolic pathways to PHBV synthesis pathway. These findings demonstrate that the CRISPRi-based gene regulation is a transformative toolkit for fine-tuning the endogenous metabolic pathways in the archaeal system, which can be applied to not only the biopolymer production but also many other applications.

3. Introduction; what is the main motivation behind the work and what is the justification for the approach adopted? There are lots of ways in which PHBV productivity can be increased, why a genetic manipulation approach? What are the advantages. At present the introduction basically says that there is an interesting technique that hasn't been applied to *H. mediterranei* before - more robust justification than this is required.

Response: Thanks for your comment. Based on your suggestion, we have included the content about the motivation behind our work, the justification for the approach adopted and the advantages of genetic manipulation (lines 67-80, and 84-85).

At present, the main limitation of PHBV application is its high production cost and low production. In order to increase the production of PHBV, a series of strategies have been adopted, such as optimizing culture conditions and feeding strategies, and using low cost agro-industrial wastes as substrate¹³⁻¹⁵. Besides, an excellent chassis cell is extremely important for the improvement of PHBV production. Genetic manipulation is a necessary means to obtain excellent chassis cells. In our previous work, exopolysaccharides (EPS) gene cluster in *H. mediterranei* was knockout to improve PHBV production by 20%¹⁶. Another engineered strain, with EPS gene cluster and phosphoenolpyruvate synthetase-like gene knockout, accumulated even higher amounts of PHBV⁴. Developing a further better production platform based on *H. mediterranei* is still ongoing and thus, more synthetic biology researches need to be conducted. Central metabolism is the main destination of carbon flux. It is difficult to direct carbon flux from central metabolism to PHBV synthesis by traditional genetic manipulation methods such as gene knockout, because that needs to deal with essential genes. In this context, a sequence-specific regulation system for gene expression is in an urgent need. Recently, CRISPRi (clustered regularly interspaced short palindromic repeats interference) is gaining considerable interests as a powerful tool for the repression of gene expression in eukaryotes and bacteria^{17,18}. Using this technique, essential genes are speculated to be repressed to various degrees and their influence on PHBV synthesis can be determined.

4. Results; This section requires a general description of the approach used and an explanation of the logic applied, in order to help guide the reader through the work. In general the results seem to be reliable and of good quality. Data are presented clearly in figures and tables and there is clearly novelty. What is the link between cell growth and PHBV production in the engineered strains? Can the authors provide an argument that it isn't just the fact that there are more cells which leads to more polymer (this doesn't seem to be the case, but this does need to be commented on).

Response: Thanks for your comment. Based on your suggestion, a general description of the approach and logic has been added in the first paragraph in the "results" section (lines 124-138).

It is difficult to apply the traditional CRISPR-dCas9 system in *H. mediterranei* for gene repression due to the high intracellular salt concentration. Thus, the endogenous

CRISPR-Cas system was used to develop the CRISPRi tool in this strain. The identification of PAM sequences is a prerequisite for the development of CRISPR-Cas-based gene editing tools. Therefore, in this study, we first identified the functional PAM sequence required for CRISPR interference in *H. mediterranei*. Next, *cas3* was knocked out to prevent the Cas3 protein from degrading target DNA. Then, different crRNAs were designed to target different regions of the key gene *crtI* involved in lycopene synthesis. The inhibitory effects generated by different crRNAs were determined. After the development of CRISPRi, the repression effects on gene expression was evaluated systematically, including different types of genes, simultaneous repression of two genes/gene clusters and the inhibitory effect of a gene in different growth periods. We subsequently applied plasmid-based CRISPRi to repress citrate synthase genes to enhance PHBV synthesis in *H. mediterranei*. Finally, the crRNA expression cassette integrated into *H. mediterranei* genome further improved PHBV productivity.

PHA production is closely related to both cell growth and PHA content. The equation describing the PHA production is as follows.

$$\text{PHA production (g/L)} = \text{PHA content (wt\%)} * \text{CDW (g/L)}$$

Cell growth is a very important factor affecting PHA production. Better cell growth usually leads to higher cell dry weight (CDW), which in turn improves the PHA accumulation. In our present study, repression of the citrate synthase genes improves both CDW and PHA content in the strain. Therefore, PHA production was significantly enhanced compared the control strain.

However, more cells do not necessarily mean more PHA production. In a study by Wu et al., the cell division of a recombinant *E. coli*, capable of producing PHB, was inhibited, thus leading to reduced cell number and elongated cell shape (Wu, H. et al. 2016, Appl. Microbiol. Biotechnol. 100, 9907-9916). These elongated cells provided larger cell volume to accommodate more PHB granules. Thus, the increased PHA content in the cells contributed to increased PHA production.

References:

Wu, H., Chen, J. & Chen, G.Q. Engineering the growth pattern and cell morphology for enhanced PHB production by *Escherichia coli*. Appl. Microbiol. Biotechnol. 100(23), 9907-9916. (2016).

5. Discussion - Add a statement about how 3HV fraction in the co-polymer is influenced by the various genetic changes made. Though there are various comments on this throughout these separate strands need to be brought together.

Response: Based on your suggestion, we have added several lines and drawn the different strands in one paragraph to discuss the 3HV fraction variation in PHBV (lines 503-507).

In contrast, the 3HV molar fraction of PHBV was reduced by 71.5% due to less 3HV monomer generated when the expression of BktB was repressed by CRISPRi tool in *H. mediterranei*. The inhibition of two citrate synthase genes *via* plasmid-based CRISPRi tool redirected more acetyl-CoA towards 3HB monomer supplying and thus reduced 3HV molar fraction.

6. Methods; generally well described and appropriate.

Response: Thanks for your positive comment.

REVIEWERS' COMMENTS:

Reviewer #1 (Remarks to the Author):

Thank you very much for your time and effort addressing all my comments and suggestions. The MS have improved significantly thanks to all the changes done based on the reviewers comments.

Reviewer #3 (Remarks to the Author):

The authors have provided reasonable responses to the issues raised in the initial review and made suitable and appropriate modifications to their manuscript. Overall the work is much improved and, in this reviewer's opinion, suitable for publication.